# Towards a comprehensive understanding of the structural dynamics of a bacterial diterpene synthase during catalysis

Ronja Driller [1], Sophie Janke[2], Monika Fuchs [3], Evelyn Warner[2], Anil R. Mhashal [4], Dan Thomas Major [4], Mathias Christmann [2], Thomas Brück [3] & Bernhard Loll [1]

Terpenes constitute the largest and structurally most diverse natural product family. Most terpenoids exhibit a stereochemically complex macrocyclic core, which is generated by C–C bond forming of aliphatic oligo-prenyl precursors. This reaction is catalysed by terpene synthases (TPSs), which are capable of chaperoning highly reactive carbocation intermediates through an enzyme-specific reaction. Due to the instability of carbocation intermediates, the proteins' structural dynamics and enzyme:substrate interactions during TPS catalysis remain elusive. Here, we present the structure of the diterpene synthase CotB2, in complex with an in crystallo cyclised abrupt reaction product and a substrate-derived diphosphate. We captured additional snapshots of the reaction to gain an overview of CotB2's catalytic mechanism. To enhance insights into catalysis, structural information is augmented with multiscale molecular dynamic simulations. Our data represent fundamental TPS structure dynamics during catalysis, which ultimately enable rational engineering towards tailored terpene macrocycles that are inaccessible by conventional chemical synthesis.

[1] Institut für Chemie und Biochemie, Strukturbiochemie, Freie Universität Berlin, Takustr. 6, 14195 Berlin, Germany. [2] Institut für Chemie und Biochemie, Organische Chemie, Freie Universität Berlin, Takustr. 3, 14195 Berlin, Germany. [3] Werner Siemens Chair of Synthetic Biotechnology, Department of Chemistry, Technical University of Munich (TUM), Lichtenbergstr. 4, 85748 Garching, Germany. [4] Department of Chemistry, Bar-Ilan University, Ramat-Gan 52900, Israel. Correspondence and requests for materials should be addressed to B.L. (email: loll@chemie.fu-berlin.de)

The intense structural diversity of terpenes is the molecular basis of diverse bioactivities, which render these compounds as development leads in the chemical, biotechnological, food and pharmaceutical industry[1,2]. Most terpenes constitute a macrocyclic core skeleton[3], which is generated by cyclisation of aliphatic oligo-prenyl diphosphates[3]. This reaction is catalysed by terpene synthases (TPSs). By contrast, chemically mimicking TPS reactions to generate tailored terpene-type macrocycles can be considered in its infancy, although promising directions have been proposed[4–9]. During the cyclisation event, the acyclic precursor has to be brought into a defined conformation that positions the leaving diphosphate group and the reactive alkene entities in proximity to initiate C–C bond forming reactions. Moreover, the substrate environment has to stabilise several propagating carbocations, to adjust to conformational changes of the substrate as well as assist in intramolecular atom transfer reactions, such as hydride or proton transfers and carbon shifts. Such enzyme guidance is necessary to control the reaction intermediates, although the inherent reactivity of carbocations is also an important component in terpene biocatalysis[10]. In addition to the complex processes in the enzyme's active site, global effects such as dynamic structural changes of the protein scaffold and inter-protein interactions during catalysis are poorly understood. A detailed understanding of dynamic, biochemical and structural effects during catalysis would ultimately enable rational engineering of this enzyme family. This capacity would be a game changing advantage in structure-activity-relation studies of

biologically relevant terpenes as even subtle structural modifications such as the additions of methyl groups to the native substrate (e.g., the "magic methyl effect")[11] or a simple H/D exchange may have a profound influence on the biological activity.

To elucidate the dynamic interactions of TPSs with their substrates during catalysis, we investigated the bacterial diterpene synthase CotB2 as a model system. This enzyme catalyses the cyclisation of the universal diterpene precursor $E,E,E$-geranylgeranyl diphosphate (GGDP) to cyclooctat-9-en-7-ol (Fig. 1a), which is subsequently elaborated to the bioactive compound cyclooctatin[12]. Cyclooctatin is a next-generation anti-inflammatory drug targeting a lysophospholipase that is upregulated in eosinophilic leucocytes rather than a cyclooxygenase inhibited by ibuprofen or aspirin[13]. The CotB2 protein sequence exhibits a modified aspartate-rich DDXD motif[12] deviating from the conventional DDXXD motif and the NSE triad NDXXSXX(R, K)(E,D) as found in bacteria and fungi[3,14], that is altered to a DTE triad in plants. Both motifs are involved in binding a conserved $Mg^{2+}$ triad that is indispensable for precise GGDP orientation in the active site. Upon substrate and $Mg^{2+}$ binding, TPSs undergo a discrete conformational change from an open, catalytically inactive one to a catalytically active closed conformation. Even though the active site cavity is already product-shaped in the open conformation, closure, accompanied by translation and rotation of secondary structure elements, is essential for catalysis. Catalysis is initiated by lysis of the GGDP

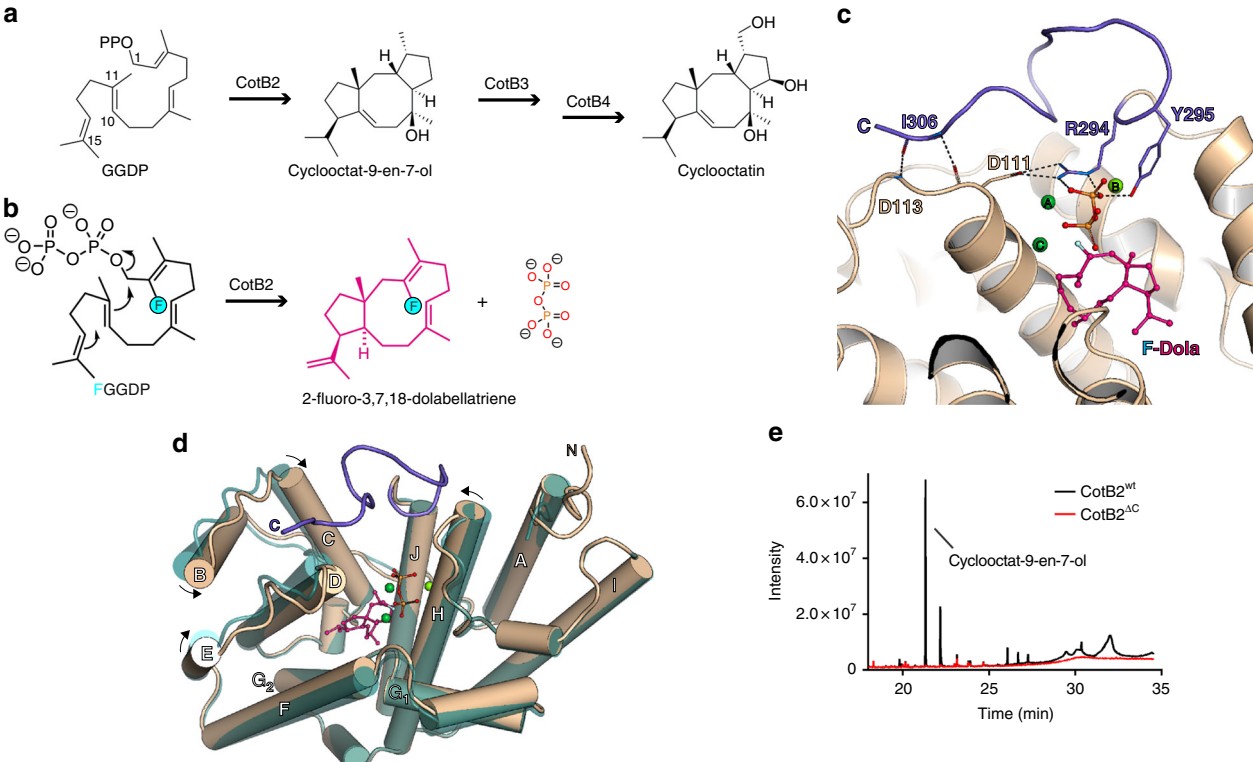

**Fig. 1** Enzymatic reaction and the structure of the closed state of CotB2, revealing the importance of its C-terminus. **a** The linear substrate geranylgeranyl diphosphate (GGDP) is cyclised by CotB2 to a fusicoccane, with a 5-8-5 fused ring system, which is subsequently elaborated to the bioactive compound cyclooctatin by two cytochrome P450 enzymes CotB3 and CotB4, respectively. **b** 2-fluorogeranylgeranyl diphosphate (FGGDP) is converted to 2-fluoro-3,17,18-dolabellatriene (F-Dola). The fluorinated position of the substrate-analogue FGGPP is indicated by a light blue circle. **c** View into the active site of $CotB2^{wt}\cdot Mg^{2+}_3\cdot$F-Dola. CotB2 is shown in cartoon representation coloured in light brown. The bound intermediate is shown in magenta and $Mg^{2+}$-ions are shown in green. Folding of the C-terminus (purple) leads to the formation of several hydrogen bonds (dashed lines), allowing for sensing of the different catalytically important motives. **d** Structural superposition of $CotB2^{wt}$ (open), shown in teal, and $CotB2\cdot Mg^{2+}_3\cdot$F-Dola (closed), shown in light brown. **e** GC/MS spectrum to monitor product formation by $CotB2^{wt}$ (black) and $CotB2^{\Delta C}$ (red). Deletion of the C-terminus results in an inactive enzyme

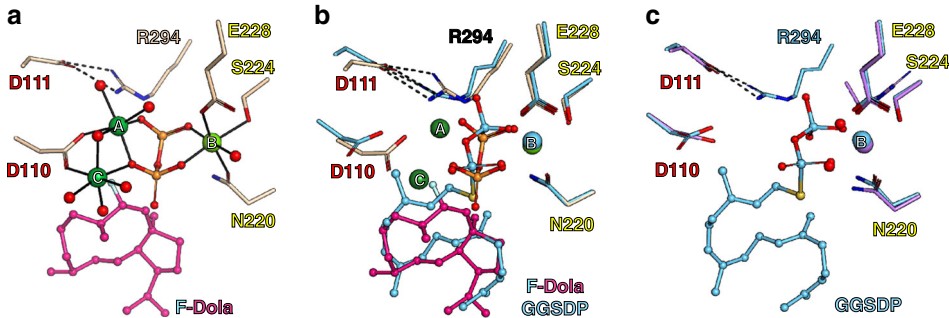

**Fig. 2** Structural comparison of different catalytic states of CotB2. Residues of the DDXD motif are labelled with red letters, residues of the NSE/DTE motif with yellow letters, respectively. Water molecules are depicted as red spheres. Hydrogen bonds are indicated by dashed lines. **a** Catalytic centre of CotB2$^{wt}$•Mg$^{2+}_3$•F-Dola. Mg$^{2+}$-ions are depicted as green spheres. Solid lines represent the coordination sphere of the Mg$^{2+}$-ions. **b** Comparison of CotB2$^{wt}$•Mg$^{2+}_3$•F-Dola shown in magenta and CotB2$^{wt}$•Mg$^{2+}_B$•GGSDP shown in cyan. **c** Comparison of CotB2$^{F107A}$•Mg$^{2+}_B$ shown in pale purple and CotB2$^{wt}$•Mg$^{2+}_B$•F-Dola shown in cyan

diphosphate moiety, a reaction mechanism characteristic for class I TPSs, generating a highly reactive, unstable carbocation[15]. This initial carbocation propagates intramolecular cyclisation, which is guided by amino acids lining the active site, where π-cation and other electrostatic interactions drive the carbocation through an enzyme-specific reaction cycle. The cyclisation cascade is terminated by lining amino acids mediated deprotonation or addition of a water molecule to the final carbocation [3,16].

Delineating interactions of lining amino acids with the substrate along the entire reaction trajectory is fundamental to understand the complexity of TPS catalysis. This information can be reliably obtained by an interdisciplinary approach combining biochemical, structural biology, computational and biocatalytic studies.

In 2014, we reported the first crystal structure of CotB2 in its open, inactive conformation without substrate. CotB2 adopts an α-helical bundle fold that is conserved among class I TPSs (PDB-ID 4OMG[17]). Single point mutations within the active site lead to pronounced structural product diversity (Supplementary Table 1 and Figs. 1, 2)[17–19].

Here, we report cumulative data that provide significant insights into the chemical mechanisms that constitute the catalytic conversion of GGDP to cyclooctat-9-en-7-ol. Remarkably, even though structural information of TPSs bound to a product or different substrate analogues as well as Mg$^{2+}$ have been reported, a complete in crystallo cyclised product has not yet been observed, demonstrating that the previously reported substrate analogues were either not correctly positioned in the active site or catalytically inactive due to chemical modifications.

### Results

**Comparison of the open and closed structure of CotB2.** In this study, we intended to probe the structural dynamics of the CotB2 reaction using X-ray crystallography of the catalytically relevant protein states in the presence of a substrate-analogue. Remarkably, we could capture the closed conformation of CotB2 by co-crystallisation with the substrate-analogue 2-fluoro GGDP (FGGDP) and determined the crystal structure at 1.8 Å (CotB2$^{wt}$•Mg$^{2+}_3$•F-Dola (2-fluoro-3,7,18-dolabellatriene); Figs. 1c, d, 2a, Supplementary Table 2). FGGDP is functionalised with a fluorine atom at the C2 position of its hydrophobic tail (Fig. 1b) that interferes with the propagation of the generated carbocation(s). A direct comparison of the open and closed conformation is possible, since our crystals are isomorphous. The overall structure of the open and closed conformation reveals significant differences. Substrate binding induces a translation and rotation of α-helices B, C, D, F and H towards the active site (Fig. 1d) to accurately

position the DDXD Mg$^{2+}$ binding motif and to bring the active site into product-shaped conformation. Upon binding of FGGDP the 12 C-terminal residues are structured to fold over the active site, resulting in the fully active and closed conformation (Fig. 1c, d). Surprisingly, we did not observe the entire FGGDP molecule. Instead, we detected clearly resolved electron density for a single diphosphate moiety and for a ring-like structure positioned in the active site that we interpret as the reaction intermediate 2-fluoro-3,7,18-dolabellatriene (F-Dola; Figs. 1b–d, 2a and Supplementary Fig. 3).

With the native substrate GGDP such in crystallo snapshots would not be feasible due to the highly unstable nature of the carbocation intermediates. Most likely the fluorinated substrate alters propagation of the carbocation and hence we do not observe the native product. The possibility of cyclisation reactions of fluorinated substrates has been demonstrated for 2-fluorofarnesyl diphosphate[20,21]. To our knowledge such a crystal structure with a trapped intermediate has never been described before for other TPSs crystallised in presence of isoprene substrates. In these reported structures, the diphosphate is bound, but not cleaved off and consequently no cyclisation reaction has occurred (Supplementary Discussion).

The diphosphate is coordinated by three Mg$^{2+}$-ions with Mg$^{2+}_B$ bound to the NSE motif and Mg$^{2+}_A$ and Mg$^{2+}_C$ being coordinated by D110 of the non-canonical DDXD motif (Fig. 2a and Supplementary Discussion and Supplementary Fig. 11). Additionally, active site residues R227 and R294 make contact with the oxygen atoms of the diphosphate moiety to compensate its negative charge. Upon binding of Mg$^{2+}_A$ and Mg$^{2+}_C$, D110 stretches out towards the active site, causing helix D movement thereby reducing the volume of the active site and trapping of the substrate. Moreover, this helix D movement positions the C-terminus for folding over the active site to shield it from bulk solvent. The closure by the C-terminus is initiated by formation of a salt-bridge between D111 and R294. In our previous studies we showed that D111E is an inactive mutant[17]. With our current data set, we can now explain this observation with the exchange at position 111 that disturbs the salt-bridge formation emphasising its catalytic relevance. Additionally, Y295 establishes hydrogen bonds to the diphosphate moiety derived from FGGDP as well as to N220 of the NSE motif. The latter RY-pair (R294/Y295) has been previously described for other TPSs[22]. Folding of the C-terminus is completed by formation of a short anti-parallel β-strand at the end of the C-terminus (Fig. 1c). Folding over of the C-terminus does not seem to be driven by strong ionic interactions, but is dominated by hydrogen bonds and van der Waals contacts.

To verify that the full closure of the active site and the folded C-terminus is not due to crystal contacts, we crystallised CotB2 with (4-amino-1-hydroxy-1-phosphonobutyl)phosphonate (alendronate (AHD); Supplementary Table 2), a compound that mimics the diphosphate group of GGDP. The CotB2$^{wt}$•Mg$^{2+}_3$•AHD structure has the same overall structure and active site architecture as described for CotB2$^{wt}$•Mg$^{2+}_3$•F-Dola, but crystallised in a different space group and clearly demonstrates that the folded C-terminus is not a crystallographic artefact caused by crystal contacts (Supplementary Fig. 4).

**Importance of C-terminal region.** In order to biochemically confirm the importance of the entire C-terminal region for catalysis, we generated a C-terminal truncation of CotB2 terminating at R294 (CotB2$^{\Delta C}$). CotB2$^{\Delta C}$ lacks the last 12 C-terminal residues, corresponding to the lid as observed in the structure of CotB2$^{wt}$•Mg$^{2+}_3$•F-Dola and CotB2$^{wt}$•Mg$^{2+}_3$•AHD. Indeed, CotB2$^{\Delta C}$ was an inactive mutant, not capable of catalysis as shown by GC/MS (Fig. 1e). Co-crystallisation of CotB2$^{\Delta C}$ with FGGDP resulted in the open-state conformation with no substrate bound (Supplementary Table 2). Additionally, this data provides further evidence that the salt-bridge between D111 and R294 is not sufficient to fully close the active site. Interestingly, there are two Mg$^{2+}$-ions bound in the CotB2$^{\Delta C}$ structure. One is fully occupied Mg$^{2+}_B$ and the other one is partially occupied Mg$^{2+}_C$, which can only be fully occupied if properly coordinated by the DDXD motif upon substrate binding and the accompanying conformational changes of the active site.

Closure of the active site seems to be an essential process in TPS catalysis. The implementation, however, appears to be very specific for each individual enzyme class. In several sesquiterpene synthases unstructured loop regions become ordered upon ligand binding, initiating active site closure[23,24]. In case of the plant-derived bornyl diphosphate synthase, a monoterpene synthase, the N-terminus together with a loop segment close the active site, and the enzyme was co-crystallised with the final product bornyl diphosphate[25]. By contrast, in the structure of the sesquiterpene aristolochene synthase from a fungus it is merely a loop segment that closes the active site[26]. Interestingly, molecular dynamic (MD) simulations of the closed state of the plant derived taxadiene synthase indicated that the N-terminus might be involved in closure of the active site[27,28]. Therefore, the mechanism of active site closure may be very specific to the reaction type and taxonomic origin of the respective TPS.

Given the importance of the C-terminus of CotB2 for catalysis, we screened the PDB for other diterpene synthases for which structural information for the open and closed conformation is available and which are structurally most related according to a DALI search[29] (Supplementary Table 3). In the closed structure of the labdane-related diterpene synthase (PDB-ID 5A0K) its C-terminus is folded as well. The RY-pair (Supplementary Fig. 5) is at the same position and engaging identical interactions with the diphosphate. Analysing protein sequences of other bacterial diterpene synthases for the presence of the RY-pair[22] and their flanking regions, we found a conserved tryptophan six amino acids upstream. Therefore, we suggest, that the WXXXXXRY ("X" any amino acid) motif is relevant for other diterpene synthases (Supplementary Fig. 5). If the tryptophan of latter motif is exchanged in CotB2 for glycine (W288G), the product becomes 3,7,18-dolabellatriene (Supplementary Table 1 and Supplementary Fig. 2)[17]. In this study, we introduced the less stereochemically demanding W288F mutation, which reduced CotB2's activity, but did not change the product profile. Thus, the aromatic character of the amino acid at position 288 in CotB2 is important for product formation and any drastic exchange to a non-aromatic side chain does interfere with the propagation of the carbocation. In the bacterial sesquiterpene synthase pentalenene synthase the corresponding mutation W308F leds to a product mixture[30]. However, in the plant-derived epi-aristolochene synthase mutations of W273 resulted in total loss of enzymatic function[23]. Therefore, tryptophan residues at this strategic position within the active site are of general importance in TPS to guide product formation.

**Pre-catalytic states of CotB2.** Very recently, Tomita et al. published a co-crystal structure of CotB2 bound to the inert substrate-analogue geranylgeranyl thiodiphosphate (GGSDP) (CotB2$^{wt}$•Mg$^{2+}_B$•GGSDP, PDB-ID 5GUE[19]), which was described as representing the closed state of CotB2. Surprisingly, the diphosphate moiety of GGSDP is coordinated merely by a single magnesium-ion (Mg$^{2+}_B$; Fig. 2c), in contradiction to the accepted principle of a trinuclear magnesium cluster required for catalysis. Notably, beside the two missing Mg$^{2+}$, even though the salt-bridge between D111 of the DDXD motif and R294 is present (Supplementary Fig. 6b), the remaining C-terminus could not be resolved[19]. From our point of view this indicates that the structure of CotB2$^{wt}$•Mg$^{2+}_B$•GGSDP represents rather a pre-catalytic state of the enzyme reflecting a snapshot from the open to the fully closed conformation (Supplementary Discussion).

To comprehend the state of the CotB2$^{wt}$•Mg$^{2+}_B$•GGSDP structure, we determined the structure of CotB2$^{F107A}$ under a Mg$^{2+}$-rich condition (CotB2$^{F107A}$•Mg$^{2+}_B$; Supplementary Table 2). Interestingly, a single magnesium-ion (Mg$^{2+}_B$) was bound to the active site, coordinated by the residues N220, S224 and E228 of the NSE motif and two water molecules (Fig. 2c and Supplementary Fig. 6a). It has been proposed for the sesquiterpene synthase aristolochene synthase, that Mg$^{2+}_B$ only binds the active site after the ligand has entered[31]. However, this does not seem to be the case for CotB2. Therefore, we suggest the binding of Mg$^{2+}_B$ to be the first step in preparing CotB2 for substrate binding.

The position of the Mg$^{2+}_B$ matches perfectly with that in the CotB2$^{wt}$•Mg$^{2+}_B$•GGSDP structure (Fig. 2c). Moreover, the conformation of the Mg$^{2+}_B$ coordinating amino acids are oriented in the same manner as there are two water molecules in our structure that mimic the oxygens of the diphosphate group in CotB2$^{wt}$•Mg$^{2+}_B$•GGSDP (Fig. 2b and Supplementary Fig. 6b). In the structure of CotB2$^{F107A}$•Mg$^{2+}_B$ the salt-bridge of D111 and R294 has not yet been established (Supplementary Fig. 6a) and the remaining C-terminus could not be resolved. This suggests, that the accurate positioning of the diphosphate moiety is required for the formation of the latter salt-bridge. The overall structure of CotB2$^{wt}$•Mg$^{2+}_B$•GGSDP is most similar to CotB2$^{wt}$ and CotB2$^{F107A}$•Mg$^{2+}_B$ as indicated by a low root-mean-square deviation (Supplementary Table 4). Hence, the structures are likely different pre-catalytic states of CotB2.

**CotB2 mechanism and structure-function dynamics.** The in crystallo capture of the reaction intermediate F-Dola provides a static snapshot along the reaction coordinate in CotB2 (Fig. 3a). The mechanism in CotB2 has previously been delineated experimentally by Meguro et al. and Sato et al., and theoretically by Hong and Tantillo[32–34]. To explain mechanistic details of the entire catalytic cascade, we turned to in silico multiscale modelling. Modelling of the CotB2 reaction commenced with the bound 3,7,18-dolabellatriene (Figs. 3b, 4). Due to the high resolution of the crystal structure that captured CotB2 in a fully closed, catalytically competent state, QM/MM modelling yielded insight into the possible role of the enzyme in stabilising the reaction intermediates. We generated all-trans GGDP in a pre-folded

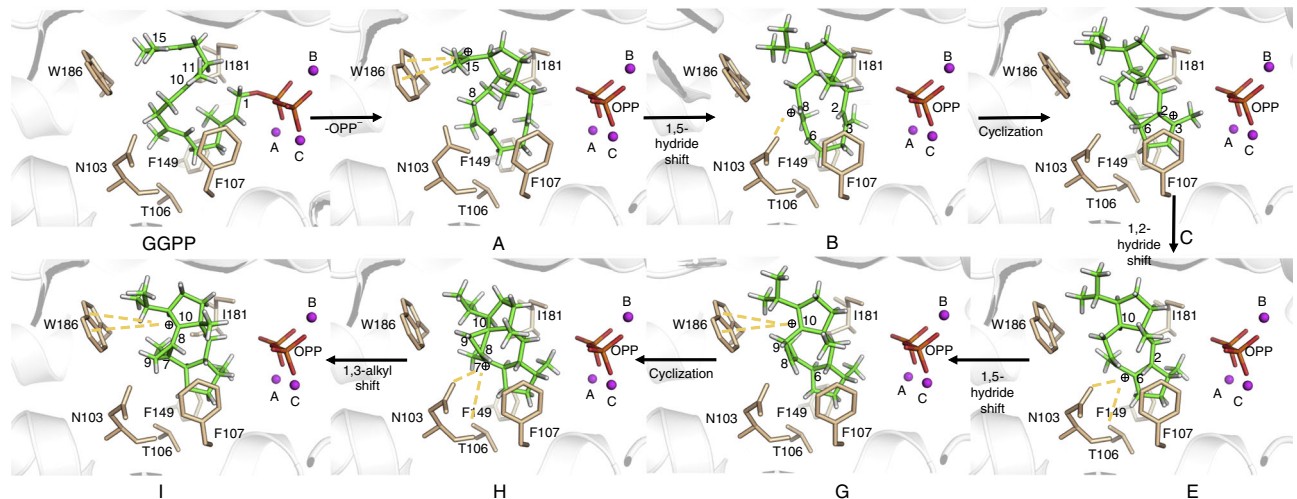

**Fig. 3** Proposed mechanism of CotB2[wt]. **a** Conversion of FGGDP to F-Dola. **b** Proposed catalytic mechanism in CotB2, as obtained in the literature from isotope labelling experiments

**Fig. 4** Mechanistic model of initial and intermediate states of CotB2. The figure shows mechanistic steps corresponding to those in Fig. 3b. The models were obtained from QM/MM simulations. The dashed yellow lines indicate interactions between the intermediate cation and CotB2

conformation, ready for catalysis, as well as all intermediate steps until the final carbocation. In the initial pre-folded GGDP state, the C1–C11 distance is 3.9 Å and the C10–C14 distance is 3.3 Å (Fig. 4). Following initial C–O cleavage and presumably concerted C1–C11 and C10–C14 bond formations, dolabellatrienyl cation (**A**) is formed, with the cation at C15 being stabilised by π-cation interactions with W186 (distance 4.6 Å; Fig. 4). The cation dislocates to C8 via a 1,5-hydride transfer reaction to yield cation **B**, as shown by deuterium labelling[32], with the cation being stabilised by an oxygen hole formed by N103 (OD1), N103 (O) and T106 (OG1) (Fig. 4).

The carbocation intermediate **B** then undergoes ring-closure via C2–C6 bond formation to yield cation **C**, with the cation located at C3 where it is stabilised by π-cation interactions, as it is

sandwiched between F107 and F149 (Fig. 4). Further stabilisation is likely provided by additional electrostatic interactions, as the cation migrates towards the diphosphate moiety during the **B→C** transformation step. Subsequently, a proposed 1,3-hydride transfer takes place to generate cation **E**. However, Hong and Tantillo[34] suggested that a series of two 1,2-hydride transfers would be energetically favoured in the gas-phase. Based on the modelled intermediate states in this study, there seems to be significant steric hindrance to a direct 1,3-hydride transfer. Therefore, two sequential 1,2-hydride transfers to provide intermediate **E** appear plausible (Fig. 3b). Identical two 1,2-hydride shifts were also experimentally reported for tsukubadiene synthase[35]. As suggested[34], these different mechanistic possibilities can be differentiated via isotope labelling as shown by Sato

et al[33]. In intermediate **E**, the cation is stabilised by F107 and the backbone carbonyl of I181 (distance 4.2 Å; Fig. 4). The following step involves another 1,5-hydride transfer to give carbocation **G**, as shown by deuterium labelling[32], with the cation located at C10, which is partly stabilised by N285 (distance 5.4 Å) and W186 (C10–W186 distance 5.3 Å; Fig. 4). A subsequent ring-closure generates a propyl-ring in intermediate **H**, while moving the cation to C7, which is stabilised by N103 (OD1), N103 (O) and T106 (OG1) (distances 3.9, 5.6, and 5.0 Å respectively; Fig. 4). The subsequent step entails a 1,3-alkyl shift to yield **I**, as established by $^{13}$C isotope labelling[32], which moves the cation back to C10 (C10–N285 distance 4.7 Å, C10–W186 distance 5.8 Å), and shifts the cyclopropyl moiety along the central octa-cyclic ring (Fig. 4). Mutation of this residue to W186H results in formation of 3,7,18-dolabellatriene[19]. This is conceptually very similar to the taxadiene synthase W753H mutation, which yields $(−)-(R)$-cembrene A[27]. The approximate energetic contribution of the above-discussed interactions between active site moieties and the carbocation intermediates were quantified using density-functional theory cluster calculations (Supplementary Tables 6 and 7). Inspection of these interaction energies clearly show that the enzyme stabilises the evolving carbocations, with values between ca. −10 and −20 kcal/mol. The reaction concludes with attack by a water molecule at position C7, followed by deprotonation, to give the final product cyclocotat-9-en-7-ol. A potential hydroxylating water molecule was not observed in the crystal structure. But based on the current mechanistic model, it could be located between N103 (OD1), N103 (O) and T106 (OG1). The role of Asn in binding hydroxylating water molecules has been stressed in the context of 1,8-cineole synthase[36]. The above mechanistic model suggests that the CotB2 active site architecture has evolved to chaperone the changing cation along the entire reaction coordinate. Future QM/MM calculations will address the complete free energy surface for the CotB2 catalysed reaction. We note that the role of the enzyme environment in guiding the carbocations along the reaction coordinate is in line with our earlier work[37–39] and that of Peters and co-workers[40,41].

## Discussion

Co-crystallising CotB2 with FGGDP enabled us to capture an intermediate state of the reaction mechanism. In combination with structural and biochemical data we could point out the importance of the properly folded C-terminus for the enzymatic activity. Our findings allowed us to propose a complete catalytic mechanism of CotB2 at a structural level by combining crystallography, biochemical tools and theoretical modelling (Supplementary Fig. 7). The enzyme mechanistic cascade commences with $Mg^{2+}_B$ binding to the NSE motif, although this binding event is not accompanied by any conformational changes of the main structural core of CotB2. Subsequently, the substrate binds with its diphosphate moiety to $Mg^{2+}_B$. This binding event induces the formation of the salt-bridge between D111 and R294. Binding of the ions $Mg^{2+}_A$ and $Mg^{2+}_C$ to the diphosphate moiety of the substrate causes the largest conformational change in the enzyme, including rigid helix motion, which reduces the active site volume, and assures proper coordination of the substrate. At this point, the complete C-terminus can fold over the active site, thereby establishing a hydrogen network between the two catalytic motifs, and shields the active site from bulk solvent. Once all the pre-catalytic binding events and conformational changes are established, GGDP is correctly positioned to subsequently undergo cyclisation, which in the case of FGGDP results in premature quenching of the cyclisation reaction, as shown by the co-crystallisation of CotB2 with FGGDP. The closure of the C-

terminus is the final trigger that initiates the chemical reaction cascade in CotB2, and possibly in additional class I TPSs.

Using theoretical modelling, we modelled the reaction using the natural substrate, GGDP. Importantly, based on in silico multiscale modelling we conclude that the three-dimensional active site architecture is such that the evolving carbocation is stabilised by precisely positioned amino acids and the bound diphosphate at each stage of the reaction cascade. Product release is possibly facilitated by the opening/unfolding of the C-terminus, which is entropically favourable, and might be the first stage in active site opening and release of the final product.

In summary, the cumulative experimental data, in conjunction with computational simulations, clearly demonstrate an active role of the protein scaffold in guiding the carbocation driven reaction towards a definitive end product. In line with our direct crystal structure evidence, there is mounting indirect evidence that the enzyme plays a crucial role in: binding the substrate, folding the substrate, selectively stabilising highly reactive carbocations to guide the reaction cascade towards the desired product and finally deprotonate or hydroxylate the final product by careful positioning of active site moieties. Therefore, the data delineated from the abrupt product trapped in the closed CotB2 structure provide fundamental advance in the understanding of TPS structural dynamics during catalysis.

Moreover, we identified universal structural features, which can be exploited for rational engineering of other bacterial TPS towards tailor-made terpene macrocycles. This information will pave the way for future semi-synthetic drug development strategies. The proof of concept for this strategy has already been well documented[42]. With respect to CotB2, site-directed mutagenesis of plasticity residues in the active site altered the enzymes product portfolio, which strongly argues for an active involvement of the protein scaffold in guiding the catalytic transformation of carbocations into distinct products[17]. The CotB2 mutation F107A afforded cembrene A as a product. The cembrene-type diterpenes display diverse bioactivities, including cytotoxic, insect deterrent and antimicrobial. Relevant work on insecticidal cembrenes has recently been reported[43]. Similarly, the single mutation of W288G alters to product to $(1R,3E,7E,11S,12S)−3,7,18$-dolabellatriene. Its structure has been reported to have potential antibiotic activity against multidrug-resistant *Staphylococcus aureus*. However, most of the targeted mutagenesis strategies were designed based on the open, non-catalytically active CotB2 structure. Therefore, the catalytic relevance of targeted amino acids in the active site could not be predicted, which often leads to generation of non-productive mutants. Moreover, one could not predict the structure of the cyclisation product. On the basis of our current model with a catalytically relevant intermediate in the active site it is feasible to trace and predict productive mutations. Furthermore, future expansion of our data to other TPSs may also enable prediction of the resulting terpene cyclisation product when targeting a specific amino acid residue(s).

## Methods

**Cloning**. DNA manipulations and cloning procedures were performed according to standard protocols (Supplementary Methods). Nucleotide sequences of all primers used in this study are summarised in Supplementary Table 5.

**Protein expression and purification**. CotB2$^{wt}$ and CotB2$^{F107A}$ fused to a C-terminal hexa-histidine-tag in a pET-24a vector were expressed in *Escherichia coli* Rosetta2 DE3 cells. Cells were harvested by centrifugation (6 min, 6000 rev min$^{-1}$ at 4 °C) and resuspended in buffer A (50 mM Tris/HCl pH 7.5, 500 mM NaCl, 5 mM MgCl₂, 1 mM DTT). The cells were lysed by homogenisation and the lysate was cleared by centrifugation (1 h, 21,000 rev min$^{-1}$ at 4 °C). Purification included Ni$^{2+}$–NTA affinity chromatography (elution in a linear gradient to buffer A containing 500 mM imidazole) and subsequent size-exclusion chromatography in buffer B (20 mM Tris/HCl pH 7.5, 150 mM NaCl, 5 mM MgCl₂, 1 mM DTT)[17,18]. CotB2$^{ΔCter}$ in pETM-11 was purified as CotB2$^{wt}$ except for an additional

proteolytic step after $Ni^{2+}$-NTA affinity chromatography. The amino-terminal $His_6$-tag of $CotB2^{\Delta Cter}$ was cleaved overnight by tobacco etch virus protease while dialysing in buffer B.

**Crystallisation**. $CotB2^{F107A}$ was concentrated to 28 mg/ml as measured by the absorbance at 280 nm. Crystals were obtained by the sitting-drop vapour-diffusion method at 18 °C with a reservoir solution composed of 17.5% (v/v) polyethylene glycol 4000, 100 mM Tris/HCl at pH 8.5 and 100 mM $MgCl_2$. For co-crystallisation experiments, $CotB2^{wt}$ was concentrated to 20 mg/ml and incubated in a fivefold molar excess of FGGDP for 30 min on ice. Crystals were obtained by the sitting-drop vapour-diffusion method at 18 °C with a reservoir solution composed of 30% (v/v) polyethylene glycol 400, 100 mM HEPES/NaOH at pH 7.5 and 200 mM $MgCl_2$. $CotB2^{\Delta Cter}$ was concentrated to 27.9 mg/ml and incubated in a 15-fold molar excess of FGGDP for 30 min on ice. Crystals were obtained by the sitting-drop vapour-diffusion method at 18 °C with a reservoir solution composed of 16% (v/v) polyethylene glycol 4000, 100 mM Tris/HCl at pH 8.5 and 100 mM $MgCl_2$. For co-crystallisation experiments with alendronate (Alfa Aesar, Germany), $CotB2^{wt}$ was incubated in a twofold molar excess for 30 min on ice. Crystals were obtained by the sitting-drop vapour-diffusion method at 18 °C with a reservoir solution composed of 26% (v/v) polyethylene glycol 4000, 100 mM Tris/HCl at pH 8.5 and 150 mM $MgCl_2$. All crystals were cryo-protected with 25% (v/v) 2-methyl-2,4-pentanediol supplemented to the reservoir resolution and subsequently flash-cooled in liquid nitrogen.

**Structure determination and refinement**. Synchrotron diffraction data were collected at the beamline 14.2 of the MX Joint Berlin laboratory at BESSY II (Berlin, Germany) and beamline P14 or beamline P11 of PETRA III (Deutsches Elektronen Synchrotron, Hamburg, Germany) at 100 K and wavelengths as followed: 0.9184 Å ($CotB2^{wt}\bullet Mg^{2+}_3\bullet$F-Dola and $CotB2^{wt}\bullet Mg^{2+}_3\bullet$AHD), 1.0332 Å ($CotB2^{\Delta C}\bullet Mg^{2+}_B$) and 0.9763 Å ($CotB2^{F107A}\bullet Mg^{2+}_B$). Diffraction data were processed with XDS[44] (Supplementary Table 2). The structures were determined by molecular replacement with the coordinates of $CotB2^{wt}$ (PDB-ID: 4OMG[17]) as search model using PHASER[45]. The structure was refined by maximum-likelihood restrained refinement in PHENIX[46,47]. Model building and water picking was performed with COOT. Geometrical restraints, used in the refinement of the fluorinated intermediate, were generated by using the PRODRG Web Server or ELBOW[48]. Model quality was evaluated with MolProbity[49] and the JCSG validation server (JCSG Quality Control Check v3.1). Secondary structure elements were assigned with DSSP[50]. Figures were prepared using PyMOL[51]. The Ramachandran plot shows that 98.64% ($CotB2^{wt}\bullet Mg^{2+}_3\bullet$F-Dola), 98.43% ($CotB2^{wt}\bullet Mg^{2+}_3\bullet$AHD), 98.73% ($CotB2^{\Delta C}\bullet Mg^{2+}_B$) and 99.64% ($CotB2^{F107A}\bullet Mg^{2+}_B$) of all residues are in favoured regions. The MolProbity Clashcores are 5.48 ($CotB2^{wt}\bullet Mg^{2+}_3\bullet$F-Dola), 4.57 ($CotB2^{wt}\bullet Mg^{2+}_3\bullet$AHD), 3.03 ($CotB2^{\Delta C}\bullet Mg^{2+}_B$) and 2.87 ($CotB2^{F107A}\bullet Mg^{2+}_B$).

**Computational modelling**. The modelling commenced with construction of cation **A** based on 2-F-3,7,18-dolabellatriene trapped during crystallisation of FGGDP in CotB2 (monomer A of PDB-ID 6GGI). The substrate GGDP was then constructed by performing the reverse reaction in silico[16]. Hydrogen atoms were added using the HBUILD facility of CHARMM. Subsequently, the enzyme was soaked in a pre-equilibrated TIP3P water[52] box of size about $80 \times 80 \times 80$ Å³, and the effect of ionic buffer environment and system charge neutralisation was accounted for by addition of 29 $Na^+$ and 20 $Cl^-$ ions[53]. Subsequently, we relaxed the system by performing step-wise energy minimisation, followed by MD simulations. The MD simulations entailed a 25 ps heating period up to 298 K followed by 5 ns of equilibration. During the first 1 ns of the equilibration simulations, weak harmonic restraints (1 kcal/mol Å²) were imposed on the backbone $C_\alpha$ atoms (residues 24–290), as well as nuclear Overhauser effect (NOE) restraints on selected hydrogen bonds in the protein and between the protein and cofactor diphosphate and $Mg^{2+}$ ions. The MD heating was performed in the NVT ensemble, while equilibration was performed in the NPT ensemble[54]. Long-range electrostatics were treated via particle-mesh Ewald summation.

Following MD equilibration, we performed hybrid QM/MM energy minimisations on the complete system, with the QM region defined as GGDP and the three $Mg^{2+}$ ions. Employing a combination of NOE restraints and general distance restraints, we generated the following cations along the reaction pathway[34]: **A, B, C, E, G, H** and **I** (Figs. 3b, 4). All MM calculations used the CHARMM protein[34] and nucleic acid force fields[55], in conjunction with parameters for GGDP[56], while QM/MM calculations[57] employed the SCCDFTB method[58] for the QM region. All simulations used the CHARMM simulations programme [59].

**Synthesis of isprenoid ligand**. Detailed experimental procedures are provided in the Supplementary Methods. Intermediate reactions steps and the final product were characterised by $^1$H-NMR (Supplementary Table 8A), $^{13}$C-NMR (Supplementary Table 8B), $^{19}$F-NMR (Supplementary Table 9A) and $^{31}$P-NMR (Supplementary Table 9B). The precise molecular weight was confirmed by mass spectrometry (Supplementary Table 10).

## Data availability

The atomic coordinates have been deposited in the Protein Data Bank with the accession code 6GGI ($CotB2^{wt}\bullet Mg^{2+}_3\bullet$F-Dola), 6GGJ ($CotB2^{wt}\bullet Mg^{2+}_3\bullet$AHD), 6GGK ($CotB2^{\Delta C}\bullet Mg^{2+}_B$) and 6GGL ($CotB2^{F107A}\bullet Mg^{2+}_B$). Other data are available from the corresponding author upon reasonable request.

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

## Acknowledgements

R. Driller is supported by Elsa-Neumann and Nüsslein-Volhard stipends. D.T. Major acknowledges support from the Israel Science Foundation (Grant #2146/15). T. Brück gratefully acknowledges funding by the Werner Siemens foundation for establishing the field of Synthetic Biotechnology at the Technical University of Munich (TUM). We accessed beamlines of the BESSY II (Berliner Elektronenspeicherring-Gesellschaft für Synchrotronstrahlung II) storage ring (Berlin, Germany) via the Joint Berlin MX-Laboratory sponsored by the Helmholtz Zentrum Berlin für Materialien und Energie, the Freie Universität Berlin, the Humboldt-Universität zu Berlin, the Max-Delbrück-Centrum and the Leibniz-Institut für Molekulare Pharmakologie. Parts of this research were carried out at PETRA III at DESY, a member of the Helmholtz Association (HGF). We would like to thank A. Burkhardt for assistance in using beamline P11 and G. Bourenkov for the assistance in using beamline P14. We are grateful to M. Wahl for continuous encouragement and support. We acknowledge support by the German Research Foundation and the Open Access Publication Fund of the Freie Universität Berlin and Technische Universität München.

## Author contributions

B.L. conceived the study. R.D. performed cloning, protein expression, purification, crystallisation, diffraction data collection and refinement. R.D. and B.L. analysed the structural data. M.F., with guidance from T.B., performed and analysed the activity assays. A.R.M. and D.T.M. performed MD calculations. S.J. and E.W. with guidance from M.C. conceived and performed organic synthesis. B.L., T.B., D.T.M. and M.C. supervised the project. R.D, B.L., T.B., D.T.M. and M.C. wrote the manuscript.

## Additional information

**Competing interests:** The authors declare no competing interests.

