## [Peer Review File · Nature Communications]

Reviewers' comments:

Reviewer #1 (Remarks to the Author):

This is an excellent manuscript, the insights into initial substrate binding are particularly noteworthy and the x-ray data is valuable. That being said, I believe it would be more appropriate for a subject-specific journal (once the following issues are addressed) since it does not describe a new method and the results are not surprising.

page 3, Despite the authors' comment about "infancy", there is actually a long, long history of chemical mimicry of terpene synthase reactions that should be mentioned.

page 3ff, The authors repeatedly talk about stabilization of carbocations. But as noted by Tantillo in a recent Angew Chem paper, most experimental data can be rationalized by "inherent reactivity" of carbocations. This should be acknowledged. While I, personally, believe that "stabilization" is likely, claims of this nature should be supported by solid energetic data from simulations, which seems to be missing in this manuscript.

page 4, The authors claim that detailed mechanistic information can only be obtained by their approach. This is an overstatement.

page 4ff, The authors claim multiple times that no cyclized products have been observed in TPS crystal structures - what about bornyl diphosphate?

page 4, Why do the authors assume that the lack of structures with cyclized substrates is a result of bad positioning or catalytic inactivity? Couldn't these structures just be weakly bound, as would be good for turnover?

page 7ff, I do not think "dimer" is the correct word. That word should be reserved for complexes of two (identical) molecules. "Pair" would be better.

page 10, "Retrosynthesis" is used incorrectly.

page 10, The authors should make it clear that the mechanism shown was worked out previously in papers 25 and 26. These papers are cited long after the mechanism is shown. In addition, why did the authors not use the published carbocation structures from these papers?

page 11, The "oxygen hole" is interesting. Could the authors provide more details and comment on its generality in TPS and other enzymes?

page 11, pi-cation interactions are just one type of electrostatic interaction, so different wording should be used.

page 12, It is inappropriate to describe an enzyme architecture as "designed"

page 14, The concluding paragraph goes too far. Why do the authors think the features they identified are "universal"? Also, this work is far from real drug development.

page 16, Why did Major not use one of his typical DFT methods here?

SI, Why are no computed structures of bound reaction intermediates included? How can readers examine these structures?

Reviewer #2 (Remarks to the Author):

This manuscript describes detailed crystallographic investigation of the bacterial diterpene synthase, CotB2, and the ensuing computational investigation of the enzymatic structure-function relationship underlying the reaction mechanism leading from the general diterpenoid precursor GGPP to the cyclooctatenol major product. The work is solid and of significant interest, although its novelty is clearly oversold. Not least in stating that this is the first structure of a TPS complexed with cyclized reaction product – e.g., in the abstract. The authors should be aware that bornyl diphosphate synthase was crystallized with its product. Indeed, the computational group (Major) has worked extensively with that TPS! This leaves the immediate impression that the authors have not bothered to familiarize themselves with the field. Frankly, this is painfully evident in their description of TPS activity, which do not catalyze “hydrolysis” (p. 4, ln 5) of the diphosphate, but rather lysis, specifically of the allylic diphosphate ester (hydrolysis would not yield a carbocation!). Crystallization of a ‘partial’ cyclized product formed from use of 2-F-GGPP provides clear insight into the ‘closed’ structure of CotB2 and a contrast to the previously reported ‘open’ (apo) structure from this group, with some differences to what now appears to be a partially closed structure from another group. The other structures reported here also help support the relevance of the fully closed structure reported here. However, while the authors have compared their structures for CotB2 to the poorly resolved structure for another bacterial diterpene synthase found in the PDB, they fail to do any comparison to the structure already reported for the bacterial ent-kaurene diterpene synthase from *Bradyrhizobium japonicum*, which would seem to be more pertinent given the much higher confidence one can have in this structure (i.e., measures of structural reliability). Finally, in their conclusion the authors again overstate the importance of their findings. While representing an important step forward in our understanding of TPS catalysis, there is no evidence here that the identified features provide any means for rational engineering of TPS activity.

A few minor clarifications should be made. First, to make the point that the AHD co-crystal provides evidence against crystallographic artefacts, it should be noted in the text of the results section that these are in a different space group, which presumably exhibits distinct crystallographic contacts (and this latter point should be clarified). Second, it should be noted that the C-terminal truncated construct that was structurally resolved is only missing the last 12 residues, corresponding to the ‘tail’ that helps ‘close’ the active site. Third, it should be noted why the F102A mutant was used for the Mg only crystallization, even if this is simply for screening purposes, as well as if there is any effect on catalysis.

Reviewer #3 (Remarks to the Author):

This manuscript summarises very interesting structural work on the diterpene synthase CotB2 for which the first crystal structure of a terpene synthase in complex with a cyclisation product was obtained. Together with the computational data important insights into the mechanism of the enzyme were obtained that deepen our understanding of diterpene biosynthesis. The manuscript is suitable for *Nat. Commun.*, but a few aspects especially about important previous contributions from other groups need to be presented with more clarity.

1. Page 3: CotB2 was first characterised by Kuzuyama and coworkers. Please cite reference 7 at the end of the sentence „This enzyme catalyses the cyclisation of the universal diterpene precursor E,E,E-geranylgeranyl diphosphate (GGDP) to cyclooctat-9-en-7-ol (Fig. 1a), which is subsequently elaborated to the bioactive compound cyclooctatin.”.
2. Page 3: The “signature aspartate-rich DDxD motif and the NSE/DTE consensus sequence” are discussed. The DDxD motif is modified here (usually DDXXD), and DTE is found in plants instead of NSE in bacteria/fungi. Please discuss in more detail.
3. Page 5, line 6: explain abbreviations “FGGDP” and “F-Dola”.
4. Page 8, line 14: ...bornyl diphosphate synthase...

5. Page 8, line 17: The sentence starting "By contrast, in the fungal..." is a bit corrupted. "aristolochene synthase" should not be capitalised.
6. Page 8, line 4 from bottom: "In the closed structure of the labdane-related diterpene synthase..." – what is the product of this enzyme? Labdane-related diterpene synthases are a large family of enzymes.
7. Page 9, line 3: Here a mutation of a tryptophan (W288G) is described. The corresponding position has previously been mutated by Cane in the pentalenene synthase (JACS 2002, 124, 7681). Please include in the discussion and cite this previous work.
8. Page 9, last paragraph: Binding of Mg^{2+} may for CotB2 be the first step, which may be different to aristolochene synthase for which it has been suggested that this Mg^{2+} only binds after the substrate has entered the active site. However, it is also possible that for both enzymes a random sequence of steps could be followed, only different snapshots have been obtained. It is also not surprising that Mg^{2+} binding is strongly favoured at high Mg^{2+} concentrations.
9. Page 11, line 5 from bottom: "Therefore, two sequential 1,2-hydride transfers to provide intermediate E appear plausible (Fig. 3b)." – Indeed the sequence of two 1,2-hydride shifts was also experimentally shown for tsukubadiene synthase that follows a very similar mechanism as for CotB2 (cf. Rabe et al., Angew. Chem. Int. Ed. 2017, 56, 2776-2779). Please include in the discussion.
10. On pages 11 + 12 the CotB2 mechanism is discussed based on the results from the quantum chemical calculations. Although Kuzuyama's labelling experiments (reference 24) are cited here, it is not explicitly mentioned that the discussed mechanism is the same as first reported by Kuzuyama. It would be fair to credit this previous work appropriately.
11. Page 13, last paragraph: The sentences "The reaction is initiated by heterolytically cleaving off the diphosphate moiety of the substrate, creating a reactive carbocation. This reactive carbocation undergoes a series of ring formations, hydride transfers, and methyl migration, before finally being quenched by a water nucleophilic attack." are very general and only repeat information that was already given in the introduction. The discussion should better focus on the findings of the present work.
12. Supporting Info, page 3: Include the NBu_4^+ cation in the name of the synthesised GGPP derivative.
13. How was the (2Z)-configuration of this synthetic compound verified? NOE?

Please note that we have change the title from “Deciphering the structural dynamics of a bacterial diterpene synthase during catalysis” to “Towards a comprehensive understanding of the structural dynamics of a bacterial diterpene synthase during catalysis “

Reviewer #1 (Remarks to the Author):

This is an excellent manuscript, the insights into initial substrate binding are particularly noteworthy and the x-ray data is valuable. That being said, I believe it would be more appropriate for a subject-specific journal (once the following issues are addressed) since it does not describe a new method and the results are not surprising.

Response:

We thank the reviewer for appreciating the quality and value of our data. However, we would like to point out that the *in crystallo* trapping of a catalytically reactive substrate intermediate in the correct orientation is a surprising result, providing that all previous attempts in the literature have failed to obtain such a structure. Moreover, the experimental data conveyed in the manuscript, in conjunction with simulations, clearly demonstrates an active role of the protein scaffold in guiding the carbocation driven reaction towards a definitive end product. Therefore, the cumulative data delineated from this structure provide fundamental advance in the understanding of TPS structural dynamics during catalysis.

page 3, Despite the authors' comment about "infancy", there is actually a long, long history of chemical mimicry of terpene synthase reactions that should be mentioned.

Response:

We thank the reviewer for this question and are delighted to comment on this issue. The reference brought at the end of the sentence in question refers to the “chemical mimicry of terpene synthase reactions” in nano-capsules. This work from the Tiefenbacher group, which we consider state-of-the-art, shows just how difficult it is to obtain chemical specificity for head-to-tail cyclization in a one-pot synthesis, as is done in enzymes (class I terpene synthases). We consider the current state of affairs as infancy compared to enzymes. This does not mean that important contributions have not been made in this area (for example McCulley et al. JACS 2017), especially for tail-to-head synthesis (mimicking class II terpene synthases). We have modified the text to reflect additional important contributions (p. 3, first paragraph):

“By contrast, chemically mimicking TPS reactions to generate tailored terpene-type macrocycles can be considered in its infancy, although promising directions have been proposed.”

We added references to papers by Pronin et al., Bartels et al., Brill et al., McCulley et al. and Zhang et al.

page 3ff, The authors repeatedly talk about stabilization of carbocations. But as noted by Tantillo in a recent Angew Chem paper, most experimental data can be rationalized by "inherent reactivity" of carbocations. This should be acknowledged. While I, personally, believe that "stabilization" is likely, claims of this nature should be supported by solid energetic data from simulations, which seems to be missing in this manuscript.

Response:

We thank the reviewer for the valuable comments as it addresses a dogmatic dispute in the terpene synthase community about the respective roles of the protein scaffold and the "inherent" carbocation reactivity in orchestrating an enzyme specific reaction cascade. With regard to the latter, the "inherent reactivity" of carbocations is no doubt an important ingredient in terpene biosynthesis. It is clear that much of terpene biosynthesis can be understood by this concept. This hypothesis, however, largely relies on theoretical calculations performed in the gas-phase. This approach per-se disregards the role of the protein scaffold in the catalytic cascade. By contrast, our crystal structure data, together with simulations, now demonstrate that the protein has a role in guiding specific folding and catalytic transformation of the intermediate carbocations towards the end-product. Our simulations, which are initiated from the experimental structure, takes the protein scaffold, as well as protein flexibility, into account and therefore goes beyond gas-phase based calculations alone. The aim of this approach is ultimately to quantify the respective contributions of the protein environment and the "inherent" carbocation reactivity on terpene synthase catalysis.

Our approach is in line with a significant amount of alternative experimental and theoretical reports (including the authors previous studies on CotB2) that indirectly demonstrate that structural alterations of the protein scaffold will alter carbocation transformation routes resulting in alternative products, which support a significant contribution of the protein scaffold on the catalytic process (see also Baer, P. et al. Angew. Chem. 2014). Additionally, Major quantified in a recent QM/MM work on mono- and sesquiterpenes the contribution of the diphosphate and selected amino acid residues on carbocation stability (ACS Catalysis 2017). This work shows a very significant effect of individual active site moieties on the free energy profiles (and hence rate) of the TPS reactions. Indeed, Major and co-workers have studied numerous terpene synthases (mono-, sesqui-, and diterpene synthases) theoretically, and our conclusion clearly points to a dominant role of the protein environment to guide carbocation transformations towards an enzyme specific end-product (e.g. JACS 2010, JACS 2012, ACS Catalysis 2017 (3 papers), Biochemistry 2018). Kindly see also

extensive experimental work from the group of R. J. Peters, which points to an active role of the enzyme and diphosphate cofactor in determining product outcome in TPS (Zhou, K. and Peters, R. J. Chem. Comm. 2011, 47, 4074; Xu, M.; Wilderman, P. R.; Peters, R. J. PNAS 2007, 104, 7397). Also, kindly see the excellent combined experimental and theoretical work coming out of the groups of Scrutton and Mullholland and co-workers, who also point to enzymatic guidance of highly reactive carbocations in TPS (for two recent papers from these groups: Leferink, N. G. H. et al. ACS Catal. 2018, 8, 3780; Karuppiah, V. et al. ACS Catal. 2017, 7, 6268).

In line with our direct crystal structure evidence, there is mounting indirect evidence that the enzyme plays a crucial role in: binding the substrate, folding the substrate, selectively stabilizing highly reactive carbocations to guide the reaction cascade towards the desired product, and finally deprotonate or hydroxylate the final product by careful positioning of active site moieties. We do not disagree regarding the crucial role of the inherent reactivity of carbocations; rather this is just one ingredient. Actually, in our opinion the terpene synthase protein scaffold and particularly the active site lining amino acids and the diphosphate moiety control this high reactivity.

In the specific case of CotB2, the role of stabilization is directly supported by our crystal structure. Due to the unique case of a bound intermediate product, it was possible to map out the entire reaction sequence using advanced QM/MM simulation techniques. Based on these calculations, it is straightforward to deduce the role of the enzyme in stabilizing the carbocation intermediates by basic physical chemistry considerations. For instance, if a crystal structure reveals a cation in proximity to a π -system, this is conventionally regarded as a π -cation interaction. This is true regardless of whether we quantified the strength of this interaction. Similarly, if a carbonyl dipole is pointing towards a carbocation, this is considered a stabilizing interaction, even if we don't quantify it. The literature is rich in such use of chemical intuition and most crystal structures are analyzed in this manner, even in cases where there is great uncertainty regarding the bound reaction intermediate states (A short list for terpene synthases: [a] Trichodiene synthase: Rynkiewicz et al. PNAS 2001, see p. 13546-13547 under section "Structure-Based Mechanism" [b] Selinadiene Synthase: Baer, P. et al. Angew. Chem. 2014, see p. 7654 [c] Bornyl diphosphate synthase: Whittington, D. A. et al. PNAS 2002, see p. 15379 under section "Mechanistic Implications"). In our case, we have both experimental evidence for the bound intermediate conformation, as well as advanced QM/MM simulations of the entire reaction cascade. Based on this we gained unparalleled insight into the role of the enzyme in stabilizing the reaction intermediates. The current simulations are based on solid, QM/MM energy and gradient calculations that, by their mathematical construction, find the intermediate's stable position in the enzyme pocket. We did not yet perform a

full-fledged QM/MM mapping of the free energy surface for this reaction. This is in progress but is beyond the scope of the current manuscript.

To quantify the strength of the current interactions and provide more “solid energetic data”, we now bring quantitative estimates of the strength of the interactions between the carbocation and the active site moieties discussed in the current manuscript. Not surprisingly, all these interactions are stabilizing, with interaction energies in the range ca. -10 to -20 kcal/mol. Future studies will address these interactions within a full-fledged free-energy QM/MM study.

In light of the important comment of the reviewer, we have modified the manuscript as follows:

1) To provide room for alternative views on enzyme catalysis, we have added the following text to the Introduction (p. 3, first paragraph):

“Such enzyme guidance is necessary to control the reaction intermediates, although the inherent reactivity of carbocations is also an important component in terpene biocatalysis.”

And we added a reference to the above-mentioned Angew. Chem. paper by Tantillo.

2) To provide quantitative support for the claims of stabilizing interactions between active site moieties and carbocation intermediates, we now provide two tables in the Supporting Information (Supplementary Table S6 and S7) with DFT based interaction energies and associated coordinates (using a hybrid DFT method that includes dispersion interactions and we also accounted for basis-set superposition error). We also modified the text in the Results section to say (p. 13, bottom of page):

“The approximate energetic contribution of the above-discussed interactions between active site moieties and the carbocation intermediates were quantified using density functional theory cluster calculations (Supplementary Table 6). Inspection of these interaction energies clearly show that the enzyme stabilizes the evolving carbocations, with values between ca. -10 and -20 kcal/mol.”

3) We also added the following sentence to clarify that complete free energy surface calculations are not included in the current work, but are forthcoming (p. 13, top of page):

“Future QM/MM calculations will address the complete free energy surface for the CotB2 catalyzed reaction.”

page 4, The authors claim that detailed mechanistic information can only be obtained by their approach. This is an overstatement.

Response:

We thank the reviewer for rightly pointing this out. Accordingly, the text was changed to (p. 4):

“This information can reliably be obtained by an interdisciplinary approach combining biochemical, structural biology as well as computational, biocatalytic studies.”

page 4ff, The authors claim multiple times that no cyclized products have been observed in TPS crystal structures - what about bornyl diphosphate?

Response:

We thank the reviewer for this comment and of course this is correct. Indeed, the crystallized product of BPPS was observed, although this is a rather unique case since the product for this reaction is bound to the diphosphate. Also, the bound substrate did not react inside the enzyme crystal. What is most novel in our work, is the capture of an intermediate that is along the reaction coordinate, and that this reaction took place in the crystal. The text has been modified to reflect this:

p. 2 (abstract, change highlighted):

“Here, we present the first structure of a TPS, the diterpene synthase CotB2, in complex with an *in crystallo* cyclised reaction intermediate and a substrate-derived diphosphate.”

p. 4 (change highlighted):

“Remarkably, even though structural information of TPSs bound to a product or different substrate analogues as well as Mg^{2+} have been reported, a complete *in crystallo* cyclised product has not yet been observed, demonstrating that the previously reported substrate analogues were either not correctly positioned in the active site or catalytically inactive due to chemical modifications.”

p. 8 (change highlighted):

“In case of the plant derived bornyl diphosphate synthase, a monoterpene synthase, the N-terminus together with a loop segment close the active site, and the enzyme was co-crystallised with the final product bornyl diphosphate.”

page 4, Why do the authors assume that the lack of structures with cyclized substrates is a result of bad positioning or catalytic inactivity? Couldn't these structures just be weakly bound, as would be good for turnover?

Response:

We thank the reviewer for this comment, and refer to relevant previous reports. If the structures were simply weakly bound, and the enzyme could readily release its product, one would assume that product release is not rate limiting. However, it has been shown in some TPS that product limits enzymatic turnover. See for example Cane, D. and Chiu, H. T. *Biochemistry* 1997, 36, 8332-8339; Mathis, J. R. et al. *Biochemistry*, 1997, 36, 8340; and see also mentioning in Zhou, K. and Peters, R. J. *Chem. Comm.* 2011, 47, 4074. Based on available experimental kinetic characterization for some

TPS, we think that product release is possibly rate limiting. The rationale for this would be the tight network of ionic interactions surrounding the diphosphate moiety, which must be broken prior to product release. So even if the terpene product is loosely bound via largely dispersion interactions, it cannot be released prior to diphosphate release. Therefore, we think that the lack of structures with cyclized substrates would be a result of bad positioning or catalytic inactivity, but we do not think it is a result of good turnover.

page 7ff, I do not think "dimer" is the correct word. That word should be reserved for complexes of two (identical) molecules. "Pair" would be better.

Response:

We thank the reviewer for pointing that out to us. We completely agree with the reviewer. We had initially chosen the term "dimer" since this motif "RY dimer" was introduced by the Dickschat group (Dickschat et al. *Beilstein J. Org. Chem.* **2017**, *13*, 1770–1780). The text has been modified to "pair" according to the reviewer's suggestion.

page 10, "Retrosynthesis" is used incorrectly.

Response:

We thank the reviewers for this comment and agree. Subsequently, the text has been modified, and now avoids using this term.

p. 10:

"Modelling of the CotB2 reaction commenced with the bound 3,7,18-dolabellatriene"

p. 15:

"The substrate GGDP was then constructed by performing the reverse reaction *in silico*"

page 10, The authors should make it clear that the mechanism shown was worked out previously in papers 25 and 26. These papers are cited long after the mechanism is shown. In addition, why did the authors not use the published carbocation structures from these papers?

Response:

We thank the reviewer for pointing this out. Accordingly, the mechanism is now mentioned earlier, and we state explicitly that the mechanism was delineated in 25 and 26. The text was changed as follows (p. 10):

"The mechanism in CotB2 has previously been delineated experimentally by Meguro et al. and Sato et al., and theoretically by Hong and Tantillo."

Appropriate references were added.

Regarding using the structures from Hong and Tantillo:

This was not considered as the starting point for the QM/MM simulations was the crystal structure, and the various intermediates were obtained by specialized restrained QM/MM minimizations inside the active site. As has been shown by Warshel et al. over the years the enzyme environment has significant effect on the reactions catalyzed. See for example the Nobel lecture in Angew. Chem. 2014, Fig. 1 and text that discussed the effect of the enzyme environment. Following in the footsteps of giants, our strategy has been to perform all enzyme calculations in the active site whenever feasible. In this particular case, the quality of the bound dolabellatriene intermediate was an excellent starting point for in-enzyme calculations. Therefore, there was no need for the published gas-phase structures of Hong and Tantillo.

page 11, The "oxygen hole" is interesting. Could the authors provide more details and comment on its generality in TPS and other enzymes?

Response:

We thank the reviewer for the comment. However, we do not have additional details, other than that this is a putative water binding site. We explicitly wrote the 3 residues we believe participate. As we state on p. 12, others have noted that an active site water binds to Asn (in cineole synthase), but did not mention additional interactions. This idea could be verified once additional crystal structures for other water-dependent terpene synthases become available.

page 11, pi-cation interactions are just one type of electrostatic interaction, so different wording should be used.

Response:

We thank the reviewer for the comment. We added the word "additional" and the new text now reads (p. 12, change highlighted):

"Further stabilisation is likely provided by additional electrostatic interactions, as the cation migrates towards the diphosphate moiety during the **B**→**C** transformation step."

On page 4, we did also spot the following sentence:

"... where π -cation and electrostatic interactions ..."

This sentence was changed to:

"... where π -cation and other electrostatic interactions..."

to emphasize that π -cation are mainly electrostatic interactions.

page 12, It is inappropriate to describe an enzyme architecture as "designed"

Response:

We thank the reviewer for pointing this out. Accordingly, the text has been changed to:

"The above mechanistic model suggests that the active site architecture in CotB2 has evolved to chaperone the changing cation along the entire reaction coordinate."

page 14, The concluding paragraph goes too far. Why do the authors think the features they identified are "universal"? Also, this work is far from real drug development.

Response:

We thank the reviewer for entering into this discussion.

As we state in the introduction, cyclooctatin, the final product of the biosynthesis cluster, is an anti-inflammatory drug targeting a lysophospholipase. We have previously described (Görner et al. *Microb Cell Fact.* 2016 May 23;15:86) the efficient heterologous reconstitution of the cyclooctatin biosynthesis cluster by introduction of a novel, non-native *Streptomyces*-derived redox system in *E. coli* that allows for high titer production of cyclooctatin. By site-directed mutagenesis of plasticity residues in the active site, we could alter the product portfolio of CotB2, which argues strongly for the active involvement of the protein scaffold in guiding the catalytic transformation of carbocations into distinct products (Janke et al. *Acta Crystallogr D Biol Crystallogr.* 2014;70(Pt 6):1528). The mutation of F107A in CotB2, yielded Cembrene A as a product. The Cembrene type diterpenes display diverse bioactivities, including cytotoxic, insect deterrent and antimicrobial compounds. Relevant work on insecticidal cembrenes has recently been published by the authors (Mischko, W. et al. *Green Chem.*, 2018, 20: 2637-2650). Similarly, the single mutation of W288G alters to product to (1*R*,3*E*,7*E*,11*S*,12*S*)-3,7,18-dolabellatriene. Its structure has been shown to have potential antibiotic activity against multidrug-resistant *Staphylococcus aureus*. Therefore, the proof of concept for generation of new bioactive molecules by altering the terpene synthase structure has been well documented by us and other authors. However, most of the targeted mutagenesis strategies were designed based on the open, non-catalytically active structure. Therefore, the catalytic relevance of targeted amino acids in the active site could not be predicted, which often lead to generation of non-productive mutants. Moreover, one cannot predict the structure of the cyclisation product. On the basis of our current model with a catalytically relevant intermediate in the active site one can trace and predict productive mutations for the first time. Moreover, future expansion of our data to other terpene cyclases may also enable prediction of the resulting terpene cyclisation product when targeting a specific amino acid residue(s).

The arguments presented here are further substantiated by data from the Keasling group, who has demonstrated the production of isoprenoid-based drugs by altering the structure of key biosynthetic enzymes, including terpene synthases (Chang MC, Keasling JD. Production of isoprenoid pharmaceuticals by engineered microbes. Nat Chem Biol. 2006;2:674).

page 16, Why did Major not use one of his typical DFT methods here?

Response:

We thank the reviewer for this comment and entering a discussion with us. We did not yet perform a full-fledged QM/MM mapping of the free energy surface for this reaction. This requires extremely time-consuming simulations. This is in progress.

SI, Why are no computed structures of bound reaction intermediates included? How can readers examine these structures?

Response:

It has not been common to provide the coordinates of modeled ligands bound to macromolecules for such studies, although this is definitely a good idea if the project is fully completed.

In our case, the theoretical part is an ongoing project. We now provide the structures of the substrate/intermediates in the active site, with surrounding amino acid moieties that are discussed in the manuscript (Supplementary Table 7). Coordinates with 3 significant digits as is standard for PDB format are provided.

Reviewer #2 (Remarks to the Author):

This manuscript describes detailed crystallographic investigation of the bacterial diterpene synthase, CotB2, and the ensuing computational investigation of the enzymatic structure-function relationship underlying the reaction mechanism leading from the general diterpenoid precursor GGPP to the cyclooctatenol major product. The work is solid and of significant interest, although it's novelty is clearly oversold. Not least in stating that this is the first structure of a TPS complexed with cyclized reaction product – e.g., in the abstract. The authors should be aware that bornyl diphosphate synthase was crystallized with its product. Indeed, the computational group (Major) has worked extensively with that TPS! This leaves the immediate impression that the authors have not bothered to familiarize themselves with the field.

Response:

We thank the reviewer for this question, which we are happy to address:

Indeed, the crystallized product of BPPS was observed, although this is a rather unique case since the product for this reaction is bound to the diphosphate. Therefore, the bound substrate did not react with the enzyme. What is most novel in our work, is the capture of an intermediate that is along the reaction coordinate, and that this reaction took place in the crystal. The text has been modified to reflect this:

p. 2 (abstract, change highlighted):

“Here, we present the first structure of a TPS, the diterpene synthase CotB2, in complex with an *in crystallo* cyclised reaction intermediate and a substrate-derived diphosphate.”

p. 4 (change highlighted):

“Remarkably, even though structural information of TPSs bound to a product or different substrate analogues as well as Mg²⁺ have been reported, a complete *in crystallo* cyclised product has not yet been observed, demonstrating that the substrate analogues were either not correctly positioned in the active site or catalytically inactive due to chemical modifications”

p. 8 (change highlighted):

“In case of bornyl diphosphate synthase from plants, a monoterpene synthase, the N-terminus together with a loop segment close the active site, and the enzyme was crystallised with the final product bornyl diphosphate.”

p. 2 (abstract, change highlighted):

“Here, we present the first structure of a TPS, the diterpene synthase CotB2, in complex with an *in crystallo* cyclised reaction intermediate and a substrate-derived diphosphate.”

Frankly, this is painfully evident in their description of TPS activity, which do not catalyze “hydrolysis” (p. 4, ln 5) of the diphosphate, but rather lysis, specifically of the allylic diphosphate ester (hydrolysis would not yield a carbocation!).

Response:

We thank the reviewer for this valuable comment. The sentence has been corrected and reads now “Catalysis is initiated by lysis of the GGDP diphosphate moiety, a reaction mechanism characteristic for class I TPSs, generating a highly reactive, unstable carbocation.”

Crystallization of a ‘partial’ cyclized product formed from use of 2-F-GGPP provides clear insight into the ‘closed’ structure of CotB2 and a contrast to the previously reported ‘open’ (apo) structure from this group, with some differences to what now appears to be a partially closed structure from another group. The other structures reported here also help support the relevance of the fully closed structure reported here. However, while the authors have compared their structures for CotB2 to the poorly resolved structure for another bacterial diterpene synthase found in the PDB, they fail to do any comparison to the structure already reported for the bacterial ent-kaurene diterpene synthase from *Bradyrhizobium japonicum*, which would seem to be more pertinent given the much higher confidence one can have in this structure (i.e., measures of structural reliability).

Response:

We appreciate the reviewers question however, we do not entirely understand what the reviewer means by “measures of structural reliability”. Based on our closed structure we conducted a DALI search for related structures to CotB2. Although ent-kaurene synthase is another bacterial diterpene structure, it is not listed on top of our DALI search and was therefore not considered for structural comparison. We have introduced a novel Supplementary Table 3, presenting the results of our DALI search. It is now clear that the labdane-related synthase LrdC is the closest structurally related diterpene synthase to CotB2.

We have changed the sentence to “Given the importance of the C-terminus of CotB2 for catalysis, we screened the PDB for other diterpene synthases for which structural information for the open and closed conformation is available and which are structurally most related according to a DALI search²⁸ (Supplementary Table 3).” From latter table, it is evident that the resolutions of the structures are sufficient to support our conclusions. Notably the crystal structures of the diterpene

synthase of *Bradyrhizobium japonicum* do not contain Mg^{2+} bound to the active site. Hence these structures were not included in our structural comparison.

Finally, in their conclusion the authors again overstate the importance of their findings. While representing an important step forward in our understanding of TPS catalysis, there is no evidence here that the identified features provide any means for rational engineering of TPS activity.

Response:

We appreciate the comment but disagree that our findings should not apply to other TPSs. For instance, the discovery that the C-terminus, including the WXXXXRY motif, is important for active site closure seems to apply also for other TPSs based on our comparison of amino acid sequences and crystal structures (see Supplementary Figure 5). Hence our findings are for instance important for the correct construct design, since secondary structure prediction programs predict the C-terminus to be unstructured. Based on our study, we can conclude that this is correct for the open state, but for the closed, active state the C-terminal portion of the protein is crucial. Too short protein constructs of other terpene synthases might hence lead to incorrect conclusions about their activity.

A few minor clarifications should be made. First, to make the point that the AHD co-crystal provides evidence against crystallographic artefacts, it should be noted in the text of the results section that these are in a different space group, which presumably exhibits distinct crystallographic contacts (and this latter point should be clarified).

Response:

We thank the reviewer for the comments. Accordingly, we have added more information explaining why the alendronate structure provides more evidence against crystallographic artefacts. The sentence reads now: "The CotB2^{wt}•Mg²⁺₃•AHD structure has the same overall structure and active site architecture as described for CotB2^{wt}•Mg²⁺₃•F-Dola, but crystallized in a different space group and clearly demonstrates that the folded C-terminus is not a crystallographic artefact caused by crystal contacts (Supplementary Fig. 4)."

Second, it should be noted that the C-terminal truncated construct that was structurally resolved is only missing the last 12 residues, corresponding to the 'tail' that helps 'close' the active site.

Response:

We appreciate the comment and accordingly added an additional sentence on page 8 providing detailed information on the CotB2^{ΔC} construct. "CotB2^{ΔC} lacks the last 12 C-terminal residues,

corresponding to the lid as observed in the structure of CotB2^{wt}•Mg²⁺₃•F-Dola and CotB2^{wt}•Mg²⁺₃•AHD.”

Third, it should be noted why the F102A mutant was used for the Mg only crystallization, even if this is simply for screening purposes, as well as if there is any effect on catalysis.

Response:

We appreciate the question and would like to elaborate on this point. Beside the single point mutation F107A as described and discussed in the framework of our manuscript, we performed identical experiments with the wild-type protein and other variants of CotB2. Most crystals were obtained in protein buffer with 5 mM MgCl₂ and additional 100 mM MgCl₂ in the crystallization buffer. Interestingly we also obtained a structure of the CotB2 F107A variant with bound Mg²⁺ under crystallization conditions without any additional MgCl₂. As the structural observations with the native protein and its variants were identical, we decided to include the structure with the highest resolution in our manuscript. Resolution is important to resolve the coordination sphere around the potentially bound Mg²⁺ ions. See also the reply to reviewer #3.

Reviewer #3 (Remarks to the Author):

This manuscript summarises very interesting structural work on the diterpene synthase CotB2 for which the first crystal structure of a terpene synthase in complex with a cyclisation product was obtained. Together with the computational data important insights into the mechanism of the enzyme were obtained that deepen our understanding of diterpene biosynthesis. The manuscript is suitable for Nat. Commun., but a few aspects especially about important previous contributions from other groups need to be presented with more clarity.

We have included in the introduction on page 2 an additional reference to a very recent publication by the Tiefenbacher and Dickschat group in Nature Catalysis (Ref 9)

1. Page 3: CotB2 was first characterised by Kuzuyama and coworkers. Please cite reference 7 at the end of the sentence „This enzyme catalyses the cyclisation of the universal diterpene precursor E,E,E-geranylgeranyl diphosphate (GGDP) to cyclooctat-9-en-7-ol (Fig. 1a), which is subsequently elaborated to the bioactive compound cyclooctatin.”.

Response:

We thank the reviewer for pointing us to the missing reference. The reference to the Kuzuyama group has been introduced at the end of the sentence on page 3, last paragraph.

2. Page 3: The “signature aspartate-rich DDXD motif and the NSE/DTE consensus sequence” are discussed. The DDXD motif is modified here (usually DDXXD), and DTE is found in plants instead of NSE in bacteria/fungi. Please discuss in more detail.

Response:

We thank the reviewer for this point. Accordingly, we have changed the sentence and provide now more detailed information about the catalytic motifs. The sentence reads now “The CotB2 protein sequence exhibits a modified aspartate-rich DDXD motif deviating from the conventional DDXXD motif and the NSE triad NDXXSXX(R,K)(E,D) as found in bacteria and fungi, that is modified to a DTE triad in plants. Both binding motifs are involved in binding a conserved Mg²⁺ cluster that is indispensable for precise GGDP orientation in the active site.”

Moreover we have included a detailed structural description of the altered aspartate-rich motif in Supplementary Discussion 2 as well as Supplementary Figure S11. A comparison of the aspartate-rich motifs' location to the epi-isozizaene synthase is presented in Supplemental Figure S11, demonstrating the effect of the degenerated aspartate-rich motif of CotB2 to the overall TPS structure.

3. Page 5, line 6: explain abbreviations “FGGDP” and “F-Dola”.

Response:

Both abbreviations are now explained in the text.

4. Page 8, line 14: ...bornyl diphosphate synthase...

The spelling mistake has been corrected.

5. Page 8, line 17: The sentence starting “By contrast, in the fungal...” is a bit corrupted.

“aristolochene synthase” should not be capitalised.

Response:

“aristolochene synthase” has been changed throughout the text. The sentence has been changed to “By contrast, in the structure of the sesquiterpene aristolochene synthase from a fungus it is merely a loop segment that closes the active site.”

6. Page 8, line 4 from bottom: “In the closed structure of the labdane-related diterpene synthase...”

– what is the product of this enzyme? Labdane-related diterpene synthases are a large family of enzymes.

Response:

We appreciate the point made by the reviewer and would like to elaborate: To our best knowledge, there is only a single publication about the labdane-related diterpene synthase (Serrano-Posada et al. *Acta Crystallogr F Struct Biol Commun.* 2015;71(Pt 9):1194), describing the protein expression, purification, and crystallization and diffraction data collection. Even though the coordinates of the structure are released, we are not aware of a publication describing its structure and function. Moreover, according to the authors, the product is not yet known. The specific labdane-related diterpene synthase is called LrdC and has only recently been identified via genome mining.

7. Page 9, line 3: Here a mutation of a tryptophan (W288G) is described. The corresponding position has previously been mutated by Cane in the pentalenene synthase (*JACS* 2002, 124, 7681). Please include in the discussion and cite this previous work.

Response:

We have included a discussion of mutational studies concerning tryptophan in TPSs from plants and bacteria. “In the bacterial sesquiterpene synthase pentalenene synthase the corresponding mutation W308F leads to a product mixture.²⁹ However in the plant-derived epi-aristolochene synthase mutations of W273 resulted in total loss of enzymatic function.²² Therefore, tryptophan

residues at this strategic position within the active site are of general importance in TPSs to guide product formation.”

8. Page 9, last paragraph: Binding of Mg²⁺+B may for CotB2 be the first step, which may be different to aristolochene synthase for which it has been suggested that this Mg²⁺ only binds after the substrate has entered the active site. However, it is also possible that for both enzymes a random sequence of steps could be followed; only different snapshots have been obtained. It is also not surprising that Mg²⁺ binding is strongly favoured at high Mg²⁺ concentrations.

Response:

We agree with reviewer that our Mg²⁺ concentrations used for crystallization are high. Please note that other groups have used identical Mg²⁺ concentrations, but do not observe Mg²⁺ bound in their structures and refer to it in the text. But one can envision different scenarios depending on sesqui- or diterpene synthases. As the reviewer suggested we cannot exclude a fully random sequence of binding events. In frame of our manuscript, we intended only to describe our observations, showing clearly a binding of a single Mg²⁺ prior to loading with the substrate, leaving the enzyme in the open, inactive state. See also the reply to reviewer #2.

9. Page 11, line 5 from bottom: “Therefore, two sequential 1,2-hydride transfers to provide intermediate E appear plausible (Fig. 3b).” – Indeed the sequence of two 1,2-hydride shifts was also experimentally shown for tsukubadiene synthase that follows a very similar mechanism as for CotB2 (cf. Rabe et al., *Angew. Chem. Int. Ed.* 2017, 56, 2776-2779). Please include in the discussion.

Response:

We thank the reviewer for the information and have therefore included an additional sentence which refers to the mechanism of the tsukubadiene synthase: “Identical two 1,2-hydride shifts were also experimentally reported for tsukubadiene synthase.”

10. On pages 11 + 12 the CotB2 mechanism is discussed based on the results from the quantum chemical calculations. Although Kuzuyama’s labelling experiments (reference 24) are cited here, it is not explicitly mentioned that the discussed mechanism is the same as first reported by Kuzuyama. It would be fair to credit this previous work appropriately.

Response:

At the beginning of this abstract we have now included a sentence referring to all mechanistic contributions: “The mechanism in CotB2 has previously been delineated experimentally by Meguro *et al.* and Sato *et al.*, and theoretically by Hong and Tantillo.”

11. Page 13, last paragraph: The sentences “The reaction is initiated by heterolytically cleaving off the diphosphate moiety of the substrate, creating a reactive carbocation. This reactive carbocation undergoes a series of ring formations, hydride transfers, and methyl migration, before finally being quenched by a water nucleophilic attack.” are very general and only repeat information that was already given in the introduction. The discussion should better focus on the findings of the present work.

Response:

The sentence has been deleted.

We have rephrased our discussion which reads now:

“The proof of concept for this strategy has already been well documented.⁴² With respect to CotB2, site-directed mutagenesis of plasticity residues in the active site altered the enzymes product portfolio, which strongly argues for an active involvement of the protein scaffold in guiding the catalytic transformation of carbocations into distinct products.¹⁷ The CotB2 mutation of F107A, afforded Cembrene A as a product. The cembrene-type diterpenes display diverse bioactivities, including cytotoxic, insect deterrent and antimicrobial compounds. Relevant work on insecticidal cembrenes has recently been reported.⁴³ Similarly, the single mutation of W288G alters to product to (1R,3E,7E,11S,12S)-3,7,18-dolabellatriene. Its structure has been reported to have potential antibiotic activity against multidrug-resistant *Staphylococcus aureus*. However, most of the targeted mutagenesis strategies were designed based on the open, non-catalytically active CotB2 structure. Therefore, the catalytic relevance of targeted amino acids in the active site could not be predicted, which often lead to generation of non-productive mutants. Moreover, one could not predict the structure of the cyclisation product. On the basis of our current model with a catalytically relevant intermediate in the active site it is feasible to trace and predict productive mutations for the first time. Moreover, future expansion of our data to other terpene synthases may also enable prediction of the resulting terpene cyclisation product when targeting a specific amino acid residue(s).”

12. Supporting Info, page 3: Include the NBu₄⁺ cation in the name of the synthesised GGPP derivative.

Response:

We acknowledge the comment and have corrected the name of the derivative to “Synthesis of (2*Z*,6*E*,10*E*)-2-Fluor-3,7,11,15-tetramethylhexadeca-2,6,10,14-tetraen-1-yl diphosphate (2-fluoro-geranylgeranyl diphosphate) tetrabutylammonium salt.”

13. How was the (2*Z*)-configuration of this synthetic compound verified? NOE?

Response:

We did not perform any NOE experiments. We separated the E/Z-isomers of geranylgeraniol by chromatography and assigned the Z-configuration by comparison to the data of Christianson et al. (Nature 2011, 469, 116-122). They proved the configuration by X-ray crystallography.

REVIEWERS' COMMENTS:

Reviewer #1 (Remarks to the Author):

I am satisfied with the authors' responses and support publication.

Reviewer #3 (Remarks to the Author):

The manuscript has been significantly improved and all the comments by the referees have been addressed satisfactorily. Reading the text again, a few more small points caught my attention that should be corrected. Also, a few new errors have been introduced.

1. The authors discuss that carbocationic intermediates towards terpenes are unstable and cannot be observed in crystal structures. At the same time, they describe their structure as a "complex with an in crystallo cyclised reaction intermediate". While I understand what the authors mean, this is not ideal, because the co-crystallised fluorinated bicyclic molecule is of course not an intermediate. More precisely, this molecule represents a derailment product of a pathway intermediate obtained with a fluorinated analog. Some passages in the abstract, last paragraph on page 5, first sentence of discussion, and first sentence on page 15 should be adjusted accordingly.

2. Page 3: "bond coupling reactions" should be rephrased, because bonds are not coupled to each other. "Bond forming" seems better.

3. Page 4: "the active site is already product-shaped" should also be rephrased, because the active site provides a template, so could be understood as a negative of the product, but has not the shape of the product.

4. Page 7, above figure 2: The last sentence is slightly corrupted. The statement is also not ideal, because it over-emphasizes the role of the substrate. At the end everything is interacting matter, to assign an active driving force to one of the participating molecules (the product), while the other molecule is passive (the enzyme) is a very subjective view.

5. Page 7, end of page: The systematic name of alendronate needs a bracket "(".

6. Page 12: hyphen missing in (-)-(R)-cembrene.

7. Page 14, last sentence of first paragraph is corrupted.

8. Page 15, top: "catalytic intermediate" should be rephrased, because there is no catalytic role of the intermediates in terpene cyclisations (the enzyme is the catalyst).

9. Page 15, middle: "cembrene A" should not be capitalised.

10. Page 15, end of discussion: the last two sentences both start with "Moreover,...".

Reviewer #3:

The manuscript has been significantly improved and all the comments by the referees have been addressed satisfactorily. Reading the text again, a few more small points caught my attention that should be corrected. Also, a few new errors have been introduced.

1. The authors discuss that carbocationic intermediates towards terpenes are instable and cannot be observed in crystal structures. At the same time, they describe their structure as a “complex with an *in crystallo* cyclised reaction intermediate”. While I understand what the authors mean, this is not ideal, because the co-crystallised fluorinated bicyclic molecule is of course not an intermediate. More precisely, this molecule represents a derailment product of a pathway intermediate obtained with a fluorinated analog. Some passages in the abstract, last paragraph on page 5, first sentence of discussion, and first sentence on page 15 should be adjusted accordingly.

We agree with the reviewer. The sentence in the abstract was modified to:

“Here, we present the structure of a TPS, the diterpene synthase CotB2, in complex with an *in crystallo* cyclised abrupt reaction product and a substrate-derived diphosphate.”

And in the discussion (p. 12):

“Therefore, the data delineated from abrupt product trapped in the closed CotB2 structure provide fundamental advance in the understanding of TPS structural dynamics during catalysis.”

2. Page 3: “bond coupling reactions” should be rephrased, because bonds are not coupled to each other. “Bond forming” seems better.

We agree with the reviewer. The sentence has been clarified and reads now: “During the cyclisation event, the acyclic precursor has to be brought into a defined conformation that positions the leaving diphosphate group and the reactive alkene entities in proximity to initiate C-C bond forming reactions.”

Since we used the same wording in the abstract, the sentence in the abstract has been corrected to “Most terpenoids exhibit a stereochemically complex macrocyclic core, which is generated by C-C bond forming of aliphatic oligo-prenyl precursors.”

3. Page 4: “the active site is already product-shaped” should also be rephrased, because the active site provides a template, so could be understood as a negative of the product, but has not the shape of the product.

We agree with the reviewer. The sentence has been modified to say:

“the active site cavity is already product-shaped”

4. Page 7, above figure 2: The last sentence is slightly corrupted. The statement is also not ideal, because it over-emphasizes the role of the substrate. At the end everything is interacting matter, to assign an active driving force to one of the participating molecules (the product), while the other molecule is passive (the enzyme) is a very subjective view.

We agree with the reviewer. The sentence was deleted.

5. Page 7, end of page: The systematic name of alendronate needs a bracket “(“.

The parenthesis has been introduced.

6. Page 12: hyphen missing in (-)-(R)-cembrene.

The hyphen has been introduced.

7. Page 14, last sentence of first paragraph is corrupted.

We are sorry for the mistake. The sentence has been corrected to “The closure of the C-terminus is the final trigger that initiates the chemical reaction cascade in CotB2, and possibly in additional class I TPSs.”

8. Page 15, top: “catalytic intermediate” should be rephrased, because there is no catalytic role of the intermediates in terpene cyclisations (the enzyme is the catalyst).

We have changed the sentence to “Therefore, the data delineated from our derailment product of a pathway intermediate trapped in the closed CotB2 structure provide fundamental advance in the understanding of TPS structural dynamics during catalysis.”

9. Page 15, middle: “cembrene A” should not be capitalised.

Typing has been corrected.

10. Page 15, end of discussion: the last two sentences both start with “Moreover,...”.

We changed the wording of the last sentence to “Furthermore, future expansion of our data to other TPSs may also enable prediction of the resulting terpene cyclisation product when targeting a specific amino acid residue(s).”